# First Insights on the Bioaccessibility and Absorption of Anthocyanins from Edible Flowers: Wild Pansy, Cosmos, and Cornflower

**DOI:** 10.3390/ph17020191

**Published:** 2024-01-31

**Authors:** Margarida Teixeira, Lorenzo De Luca, Ana Faria, Matteo Bordiga, Victor de Freitas, Nuno Mateus, Hélder Oliveira

**Affiliations:** 1REQUIMTE/LAQV, Chemistry and Biochemistry Department, Faculty of Sciences, University of Porto, 4169-007 Porto, Portugal; up202103465@edu.fc.up.pt (M.T.); vfreitas@fc.up.pt (V.d.F.); nbmateus@fc.up.pt (N.M.); 2Department of Pharmaceutical Sciences, Università del Piemonte Orientale, Largo Donegani 2, 28100 Novara, Italy; 20027861@studenti.uniupo.it (L.D.L.); matteo.bordiga@uniupo.it (M.B.); 3CHRC, NOVA Medical School (NMS), Faculdade de Ciências Médicas (FCM), Universidade NOVA de Lisboa, 1169-056 Lisboa, Portugal; ana.faria@nms.unl.pt; 4CINTESIS@RISE, NOVA Medical School (NMS), Faculdade de Ciências Médicas (FCM), Universidade NOVA de Lisboa, 1169-056 Lisboa, Portugal

**Keywords:** anthocyanins, bioaccessibility, edible flowers, food processing, intestinal absorption

## Abstract

Edible flowers are regaining interest among both the scientific community and the general population, not only for their appealing sensorial characteristics but also from the growing evidence about their health benefits. Among edible flowers, those that contain anthocyanins are among the most consumed worldwide. However, little is known regarding the bioaccessibility and absorption of their bioactive compounds upon ingestion. The aim of this work was to explore, for the first time, the behavior of anthocyanin-rich extracts from selected edible flowers under different food processing conditions and after ingestion using simulated digestions, as well as their absorption at the intestinal level. Overall, the results showed that the monoglucoside and rutinoside anthocyanin extracts were less stable under different pH, temperature, and time conditions as well as different digestive processes in the gastrointestinal tract. There was a prominent decrease in the free anthocyanin content after the intestinal phase, which was more pronounced for the rutinoside anthocyanin extract (78.41% decrease from the oral phase). In contrast, diglucoside and rutinoside anthocyanin extracts showed the highest absorption efficiencies at the intestinal level, of approximately 5% after 4 h of experiment. Altogether, the current results emphasize the influence of anthocyanins’ structural arrangement on both their chemical stability as well as their intestinal absorption. These results bring the first insights about the bioaccessibility and absorption of anthocyanins from wild pansy, cosmos, and cornflower and the potential outcomes of such alternative food sources.

## 1. Introduction

Edible flowers have been part of human culinary practices since ancient times, adding a touch of elegance and flavor to various dishes. Across cultures, these vibrant blooms were not only appreciated for their aesthetic appeal but were also integrated into traditional diets for their potential health benefits [1]. In recent years, there has been a resurgence of interest in edible flowers within both the scientific community and the general population. This growing fascination is driven by a desire to explore alternative food sources, delve into the nutritional and health-promoting properties of botanicals, and create a more diverse and sustainable food culture. Edible flowers are now finding their way onto the plates of fine-dining restaurants, home kitchens, and experimental food laboratories. In fact, edible flowers are not just a feast for the eyes; they also offer a range of potential health benefits. Many flowers contain bioactive compounds, including vitamins, minerals, and antioxidants like polyphenols, which contribute to their nutritional value. Additionally, certain flowers have been traditionally used for their anti-inflammatory, antimicrobial, and mood-enhancing properties [2]. Among the diverse array of edible flowers, those rich in anthocyanins have garnered special attention [3,4].

Anthocyanins correspond to the main group of water-soluble secondary metabolites that are produced by plants, and that are mainly known for the variety of colors that they give to the plants [5,6]. Their name derives from the ancient Greek words “anthos” and “kyaneos”, which mean flower and dark blue, respectively [7,8]. Anthocyanins are mostly found in their glycosylated form, being that their structure derives from the flavylium cation, in which hydroxyl or methoxy groups can bind to its aromatic rings (Figure 1) [9,10].

There are 17 naturally occurring anthocyanidins; however, edible flowers typically contain delphinidin, peonidin, pelargonidin, malvidin, cyanidin, and petunidin [11,12]. In natural fonts, an abundance of non-methylated anthocyanidins, which include cyanidin (30%), delphinidin (22%), and pelargonidin (18%), has been generally noticed, leaving malvidin (7.5%), peonidin (7.5%), and petunidin (5%), the methylated anthocyanins, with lower percentages of abundance [8,13]. Presently, about 700 anthocyanins have been described in several natural sources, and only 200 of them have been properly characterized [10,14].

These compounds are linked to various health advantages, such as providing antioxidant and anti-inflammatory effects, safeguarding cardiovascular health, and potentially possessing anti-cancer properties [14,15,16,17,18].

From the vast diversity of anthocyanin-rich edible flowers, cornflower (*Centaurea cyanus*), wild pansy (*Viola tricolor*), and cosmos (*Cosmos bipinnatus*) are among the most consumed species worldwide (Figure 2) [1,4].

Despite the growing interest in these types of edible flowers, there is still much to uncover regarding their consumption outcomes. To fully understand the biological relevance, that is, the health benefits associated with the consumption of anthocyanin-rich edible flowers, prior knowledge about the stability, bioaccessibility, and bioavailability of these bioactive compounds is required [14,19]. Anthocyanin stability comprises the changes (or protection to some extent) of their physicochemical properties throughout the digestive processes along the gastrointestinal tract (GIT) [20]. The release of anthocyanins from plant cell vacuoles is a critical step before their absorption as bioactive compounds. Food processing methods, along with the act of food chewing, can promote an increase in the free anthocyanin content of these food sources, thus impacting the nutritional content and nutraceutical aspect of diets [21,22]. On the other hand, anthocyanin bioavailability is a broad concept that is usually defined as the fraction of digested anthocyanins that enters the systemic circulation, making them available for a range of target tissues and organs where they exert their biological effects [20,23]. Anthocyanins normally exhibit low bioavailability. The reported bioavailability of anthocyanins ranges from 2% to less than 1%, indicating that only a small fraction effectively reaches the target tissues [20,22,24,25,26,27,28]. The low bioavailability values are a result of the high conjugation, colon microbiota metabolism, and excretion of the ingested anthocyanins through urine and feces [21]. However, assessing the bioavailability of food bioactive compounds is still very challenging since it englobes their bioaccessibility, their absorption, their biotransformation, and bioactivity altogether [19,23]. Also, the majority of the studies focus on the same type and sources of anthocyanins, which may also underestimate the overall conclusions regarding such issues [14].

In the case of cornflower, wild pansy, and cosmos, there is no knowledge about the fate of their anthocyanins when submitted to different conditions during food processing and upon their consumption.

Therefore, in this study, we focused our efforts on understanding the effect of time, pH, and temperature on the anthocyanin content of the selected edible flowers as well as understanding their fate after ingestion at the upper gastrointestinal tract using in vitro approaches such as simulated digestions and the evaluation of their absorption using intestinal cell models.

## 2. Results and Discussion

### 2.1. Anthocyanin Extraction and Purification from the Edible Flowers

In order to obtain pure extracts of anthocyanins from each edible flower, a multistep procedure was applied to the water-extracted resulting solutions. This enabled isolation of anthocyanins from all the other water-soluble compounds present in the samples. In fact, although traditional approaches to obtaining anthocyanin-rich extracts from different food sources such as liquid-liquid extraction with an organic phase (typically ethyl acetate) and water phase followed by a solid-phase extraction with reversed-phase C_18_ to remove undesired compounds such as sugars, small acids, phenolics, and amino acids from the extracts are widely applied due to their rapid and low-cost procedures, in many cases the purity degree of the extract is not as high as desirable and they usually require large amounts of organic solvents [29,30]. In particular, some polyphenols are extremely difficult to isolate from anthocyanins using such traditional approaches, namely, flavonol glycosides due to their structural similarities. Such issues can impact the outcomes and the interpretation of different results focused on anthocyanin bioactivity and bioavailability. Therefore, more refined methodologies must be applied to get purified anthocyanin extracts. Cation-exchange chromatography has emerged as an alternative to separate charged compounds from non-charged compounds. In the case of anthocyanins, as it is well known, they can adopt different charged states depending on the pH of the solution, which not only alters their structural features but also their interactions with the stationary phases [6]. Therefore, cation-exchange chromatography used as a solid-phase extraction methodology can represent a valuable tool to obtain purified anthocyanin extracts. Anthocyanins can be converted to positively charged flavylium cations in acidic conditions. Such cation structures are retained on the negatively charged sulfonic acid groups of the resins through ionic interactions. However, flavonol glycosides and phenolic acids are predominantly neutral (protonated) under acidic conditions due to their higher pK_a_ [31]. Therefore, these compounds are not likely to be absorbed by the resin and are eluted by the solvent, while anthocyanins remain retained in such conditions. When raising the pH, anthocyanins will get either a neutral or negative charge and will be further eluted.

In fact, recent studies have shown the potential of this tool to isolate anthocyanins from different food sources. Liao et al. have successfully isolated anthocyanins from mulberry using a cation-exchange resin 001X7 and obtained a purification degree of about 95% [32]. On the other hand, a comparison between strong cation-exchange (SCX) and C_18_ sorbents was applied to separate anthocyanins from red cabbage [33]. The authors concluded that, in the case of the cation-exchange methodology, a buffer with pH 6 and methanol were necessary to elute anthocyanins from the resin. Another study compared the purification degree of anthocyanins from different food sources using different techniques and concluded that the mixed-mode cation-exchange chromatography (Oasis^®^ MCX sorbent that features combined mechanisms of strong cation-exchange and reversed-phase adsorption) drastically increased the purity of the extracts when compared to the other techniques applied [34].

Due to the fact that the selected edible flowers for this study are rich in flavonol glycosides in addition to anthocyanins [4], a mixed-mode cation-exchange Discovery^®^ DSC-MCAX was utilized after the application of the traditional methodologies such as liquid-liquid extraction and reversed-phase C_18_. DSC-MCAX consists of a packed bed that contains both octyl (C8) and benzene sulfonic acid (SCX) bondings allowing a double retention mechanism based on reversed-phase and cation-exchange. After the application of the extracts in acidic conditions (at which the anthocyanins are in their flavylium cation form with a positive charge being retained by the resin functional groups), the non-anthocyanin compounds were eluted with acidic methanol, as they do not present the same particularity of changing their charge according to the pH, therefore exhibiting a net charge of zero. Then, the pH of the system was elevated with the phosphate buffer to pH 6 and, with the presence of methanol, the anthocyanins were further eluted. At this pH value, anthocyanins are within solution in a dynamic network of different structures, with the majority of them in the hemiketal pseudobase and quinoidal base structural forms, acquiring an overall negative charge; therefore, the functional sulfonic groups of the resin no longer establish a stable linkage with them. The results showed that the obtained fractions got an outstanding purity degree for all the edible flowers (Figure 3). The chromatograms recorded at 280 nm showed that the detected compounds are mainly anthocyanins (based on the diode array detector and the mass spectrometry analyses—see Section 2.2), and the presence of compounds other than anthocyanins were undetectable, confirming the high purity degree of the different extracts.

### 2.2. Anthocyanin Characterization of the Edible Flowers’ Purified Extracts

#### 2.2.1. Wild Pansy (*Viola tricolor*)

LC-MS analysis on positive mode enabled identification of seven anthocyanins present in the extract of wild pansy—[M-H^+^] at *m*/*z* 919.18, 773.13, 919.33, 757.18, 1081.27, 903.36, and 757.34, which was further confirmed by the typical absorption maxima around 280 and 520 nm.

It is possible to observe that all the anthocyanins present a rutinoside moiety in their core structure (292 m.u.). In previous studies, the same was observed for the Viola tricolor L. species, although the main anthocyanins were not the same as those herein, which may be explained by intraspecies differences [35]. Also, in other species from the Viola family, like in the case of Viola odorata and Viola x wittrockiana, the same type of anthocyanins were found [36,37].

In the present case, the main anthocyanin present in the extract was compound **3** (Figure 3a and Table 1) with a parent ion [M-H]^+^ at *m*/*z* 919.33. The fragmentation profile with product ions at *m*/*z* 757.25, *m*/*z* 465.13, and *m*/*z* 303.12 might correspond to the loss of a p-coumaric acid moiety (162 m.u.) followed by the loss of a rutinoside moiety (292 m.u.) and the loss of a glucose moiety (162 m.u.). The ion at *m*/*z* 303.12 is compatible with the aglycone of delphinidin. Based on this and on the loss patterns, this compound was tentatively classified as Delphinidin-3-(4′-trans-p-coumaroyl)-O-rutinoside-5-O-glucoside. The cis isomer was eluted earlier at 4.41 min. Compound **5** (Figure 3a and Table 1) with a parent ion [M-H]^+^ at *m*/*z* 1081.27 was tentatively attributed to Delphinidin-3-(4′-trans-p-coumaroyl)-O-rutinoside-5-O-(6′-caffeoyl)-glucoside. The first product ion at *m*/*z* 756.80 corresponds to a loss of 324 m.u., which together with the UV spectrum is compatible with the loss of caffeoyl-glucoside moiety [38]. All the other compounds were tentatively identified based on the same principles.

#### 2.2.2. Cosmos (*Cosmos bipinnatus*)

In the case of cosmos, LC-MS analysis on positive mode enabled identification of five anthocyanins present in the purified extract. Compounds **2** and **4** (Figure 3b and Table 2) were the main ones present in the extract with parent ions [M-H]^+^ at *m*/*z* 595.32 and [M-H]^+^ at *m*/*z* 609.28, respectively. The fragmentation product ions for compound **2** at *m*/*z* 449.17 and at *m*/*z* 287.08 corresponded to losses of 146 and 164 m.u., which indicate the potential loss of a p-coumaric acid followed by the loss of a glucoside moiety. The smallest product ion at *m*/*z* 287.08 is compatible with the presence of the cyanidin aglycone. Therefore, this compound was tentatively classified as Cyanidin 3-O-(6″-p-coumaroyl-glucoside). On the other hand, the fragmentation product ions for compound **4** at *m*/*z* 463.18 and *m*/*z* 301.11 corresponded to losses of 146 and 164 m.u., as in the previous case, therefore indicating the presence of a p-coumaric acid and a glucoside moiety. However, in this case, the smallest product ion at *m*/*z* 301.11 is compatible with the aglycone of peonidin. Therefore, this compound was tentatively classified as Peonidin 3-O-(6″-p-coumaroyl-glucoside).

#### 2.2.3. Cornflower (*Centaurea cyanus*)

In the purified extract of cornflower, it was possible to identify eight anthocyanins. Most of the anthocyanins were derived from the aglycone of cyanidin, which was confirmed by the presence of a fragmentation product ion at *m*/*z* 287 in all compounds except in the case of compound **7**. Interestingly, a previous study was able to identify three main anthocyanins in *Centaurea cyanus* L. that correspond to the main anthocyanins identified herein—compounds **1**, **4**, and **6** (Figure 3c and Table 3). Compound **1** with a parent ion [M-H]^+^ at *m*/*z* 611.31 and fragmentation product ions at *m*/*z* 449.19 and *m*/*z* 287.15, which correspond to the loss of 162 m.u. in each, is compatible to the consecutive loss of two glucoside moieties. Therefore, this compound was tentatively classified as Cyanidin-3,5-O-diglucoside. Compound **4** presents a parent ion [M-H]^+^ at *m*/*z* 679.34 and fragmentation product ions at *m*/*z* 535.25 and *m*/*z* 287.15, which correspond to losses of 162 m.u. and 248 m.u., respectively. This indicates the presence of a glucoside moiety attached to one carbon of the flavylium structure and the presence of a malonylglucoside moiety attached to another carbon. Therefore, this compound was tentatively identified as Cyanidin-3-O-(6″-malonylglucoside)-5-O-glucoside. It is important to notice that compound **3** presented the same characteristics as compound 4 in terms of parent and products ions, which suggested the presence of two isomers of the Cyanidin-3-O-(6″-malonylglucoside)-5-O-glucoside, a cis-like isomer and a trans-like isomer, potentially of the malonic acid substituent; however, it was not possible to reach a conclusion about the elution order between both. Compound **6** presented a parent ion [M-H]^+^ at *m*/*z* 711.38 with fragmentation product ions at *m*/*z* 549.23 and *m*/*z* 287.15, which corresponded to losses of 162 m.u. and 262 m.u., respectively. This fragmentation pattern is compatible with the presence of a glucoside and a succinyl glucoside moiety. Therefore, this compound was tentatively classified as Cyanidin-3-O-(6″-succinylglucoside)-5-O-glucoside.

### 2.3. Bioaccessibility of the Anthocyanins from Wild Pansy, Cosmos, and Cornflower

#### 2.3.1. Temperature and pH

Temperature and pH were the variables chosen to be tested throughout these assays as they are known for affecting both anthocyanin color and chemical stability. Generally, higher temperatures are associated with a lower anthocyanin stability and consequently a lower bioaccessibility. Moreover, a continuous exposure to these conditions may alter the anthocyanin content, leading to changes in color and antioxidant activity. This is due to a higher susceptibility of the anthocyanin structure to the nucleophilic attack by water, thus leading to changes in its structural arrangement [39]. Figure 4 shows the effect of both temperature and pH, over time, on the anthocyanin-rich extracts of the selected edible flowers.

Overall, it is possible to observe in Figure 4 that the pH had a negative effect on the bioaccessibility of the anthocyanins present in the different extracts. In the case of wild pansy, both temperature and time were crucial for the negative effects of pH. Wild pansy anthocyanins maintained a good stability up to pH 7 at low temperatures (Figure 4a) during the entire experimental time; however, the stability started to decrease rapidly with the increase in temperature from 60 °C up to 100 °C. This was more pronounced for pH values 5–11 (Figure 4b–d). At 100 °C, regardless of the pH, the anthocyanins bioaccessibility decreased to values lower than 50% after only 15 min, revealing a high susceptibility of wild pansy anthocyanins to such temperatures (Figure 4d). Interestingly, pH 3 showed the highest variation of behavior, which may be related to the dynamic network of structures in solution at this particular condition (neutral hemiketal pseudobase and flavylium cation).

In the case of cosmos, the results followed the same trend seen in the wild pansy, with temperature acting as a potentiator of the negative effects of pH over time. It is possible to notice, however, that compared to wild pansy, the extract of cosmos was stable up to pH 7 at temperatures as high as 80 °C (Figure 4e–g) with more than 50% of the total anthocyanins detected after each time period and, below that, a good stability was maintained up to pH 9 during the entire experimental time (Figure 4e,f). At 100 °C, the anthocyanins from cosmos were able to maintain their stability for the lower pH values, once again, where the predominant species are neutral hemiketal pseudobase and flavylium cation (Figure 4h).

Concerning cornflower, once again, the trend was similar to the previous two edible flowers. However, it is possible to observe that cornflower anthocyanins were consistently more stable (and therefore bioaccessible) under the different conditions tested when compared to the others’ anthocyanins, with an exception in the case of 60 °C (Figure 4j).

From the results represented in Figure 4, it is clear that lower pH values are more favorable to the bioaccessibility of the anthocyanins. This can be explained by the well-known dynamic structural network that anthocyanins acquire in solution depending on the pH. The flavylium cation is the most stable structure and predominant at low pH values. At higher pH values anthocyanins tend to suffer two types of reactions that originate new structures: a proton-transfer reaction and a nucleophilic attack by water molecules within the pyran ring. These reactions result in the appearance of quinoidal bases and hemiketal forms (the latter being predominant at slightly acidic pH values) [14]. However, the parent structure of the anthocyanins is shown here to have an important role in stability at different pH values. For instance, in the case of cosmos we predominantly have acylated monoglycosides, whereas in the case of cornflower the main anthocyanins are acylated polyglycosides, as presented in Table 2 and Table 3, respectively. Previous studies have shown that polyglycosylated anthocyanins tend to be more stable than their monoglycosylated counterparts [14]. In fact, it was found that the hydrated forms of anthocyanins (hemiketal) are more stable in the case of 3,5-O-diglucosides than in 3-O-glucosides [40]. Interestingly, when comparing the results of cosmos and cornflower, it is possible to observe a higher stability of both hemiketal and quinoidal forms (pH 5–11) in the latter. This suggests that a higher number of sugar units (and acyl groups) is potentially capable of establishing a higher number of intermolecular interactions, shielding the structure from degradation.

Interestingly, especially for higher pH values, the anthocyanins from wild pansy that were identified as being mainly polyglycosylated (Table 1) showed a lower stability/bioaccessibility when compared to the other sources. Moreover, wild pansy is the only edible flower that exhibits anthocyanins glycosylated with rutinoside, a disaccharide constituted by a glucose and a rhamnose. The lack of a hydrogen bond donor in C6 of the rhamnose unit of the rutinoside may compromise the formation of intramolecular hydrogen bonds, which help stabilize the anthocyanin structure [41]. Such phenomena may explain the results obtained in the case of wild pansy for higher pH values when compared to the other anthocyanin sources.

These results provide important insights into the behavior of each edible flower’s anthocyanin extracts when submitted to distinct cooking techniques and/or food processing methods. Such knowledge is crucial when thinking about the inclusion of edible flowers in different food preparations and, consequently, in the daily diet. Therefore, it is imperative to expand our knowledge regarding the stability of the main anthocyanins of each edible flower to understand how to cook or process them in order to have the greatest phytochemical outcome.

#### 2.3.2. Simulated Digestions

The simulated digestions were performed according to a static in vitro digestion model stipulated by INFOGEST, used to simulate digestion in the most important cavities for food digestion: the mouth, stomach, and small intestine. This work aimed to see if the digestion processes interfere with edible flowers’ anthocyanin content, prior to their absorption. The results obtained are represented in Figure 5.

Overall, the free anthocyanin content detected was not heavily affected during the oral phase. For wild pansy and cosmos, the free anthocyanin content increased during the oral phase (9.3% and 12.5%, respectively), which may be a result of the disaggregation of anthocyanin complexes, increasing their bioaccessibility without compromising their stability. The conditions at which the oral digestion is performed, highlighting the neutral pH, make anthocyanins more prone to degradation and amylase (the main enzyme in the oral cavity) also plays a part in anthocyanin degradation. However, this degradation is described as being a time-dependent process and since the oral phase lasted for ten minutes it was not able to affect the edible flowers’ anthocyanin stability to a large extent. Moreover, the sugar moieties in the anthocyanins are important factors for their structural stability [42].

In the gastric phase, there was an overall decrease in the free anthocyanin content when compared to that detected in the oral phase, and it was more pronounced for the wild pansy (with a decrease of 61.86% from the oral phase). Nevertheless, in several studies an increase in the polyphenolic bioaccessibility has been reported during the gastric phase as it is performed at pH 3 [43,44,45]. Even though at this pH value anthocyanins are in their most stable structural form, the presence of lipase, pepsin, and gastric fluids throughout the three hours of the gastric digestion may greatly affect their chemical stability, justifying the overall decrease in the free anthocyanin content and, thus, decreasing its bioaccessibility at this digestion step.

In the intestinal phase, there was a general decrease in the free anthocyanin content when compared to that detected in the gastric phase, once again more pronounced for the wild pansy extract (with a decrease of 78.41% and 16.55% from the oral and gastric phases, respectively). The characteristic rutinoside sugar unit may impair the inter- and intramolecular interactions that the main anthocyanins from this edible flower can establish, thus greatly compromising its stability. The conditions used to simulate the small intestinal compartment (pH 7) make the anthocyanins transition from their flavylium cation form predominantly to their quinoidal base form, which is stated as highly unstable. Overall, there is a pronounced decrease in the free anthocyanin content after the intestinal phase, in accordance with what is described in other studies with different anthocyanin sources [46,47].

### 2.4. Cytotoxicity Assays

Prior to the transepithelial transport experiments, an evaluation of the cytotoxicity of the edible flowers’ purified anthocyanin extracts on a Caco-2 cell line was conducted. The information obtained is of high importance to ensure that the next experiments are not compromised by the anthocyanin extract concentration interfering with the viability of the cell lines and potentially compromising the integrity of the cell barriers formed. This evaluation assumes a critical significance for establishing an optimal anthocyanin concentration for subsequent transepithelial absorption experiments. Ideally, the anthocyanin concentration should be selected based on the concentration at which cell viability is maximally preserved, thus ensuring the integrity of the cell monolayers. With this aim, an MTT assay was performed and the results are represented in Figure 6.

Overall, the concentration-response curve pertaining to the Caco-2 cells showed that each different extract presented different cytotoxicity levels towards this cell line. As none of the edible flower extracts exhibited substantial cytotoxicity within the tested concentration range, a concentration of 0.8 mg/mL was selected for conducting the transepithelial absorption experiments.

### 2.5. Transepithelial Absorption Assays

Transepithelial absorption experiments were conducted at the intestinal level using the Caco-2 cell line. Purified anthocyanin extracts and their corresponding intestinal digestions were applied to the apical surface of the cell monolayers cultured on Transwell^®^ permeable inserts at a final concentration of 0.8 mg/mL. The intestinal digests were previously diluted to match the concentration of the anthocyanin extract solutions and to ensure no cytotoxicity related with the presence of the enzymatic system utilized in the digestion processes [48]. Subsequently, anthocyanin transport across cell barriers was monitored over 4 h. For this, several aliquots were collected from the basolateral side at different time points to evaluate the anthocyanin transport efficiency over time. Only parental anthocyanins were detected in the basolateral compartment at both 280 and 520 nm, suggesting that no apparent metabolization occurred during the experiment period.

The results obtained for the transepithelial absorption of the anthocyanin extracts and their respective intestinal digestions in the Caco-2 monoculture are shown in Figure 7.

In both absorption experiments, the transepithelial transport was time dependent, increasing over the course of 4 h in distinct ways for every edible flower. A time dependent transport was also observed in other studies with structurally complex anthocyanin sources such as purple sweet fleshed potato (PSFP) [22]. No significant individual differences were found among the anthocyanins present in the extracts; therefore, the analysis was based on the concentration of the total peaks in each sample. Moreover, the results obtained for the simulated digestions indicated a significant reduction in the free anthocyanin content after the intestinal phase. In the transepithelial absorption assays, there were no substantial differences in the transport efficiency values between the assays with purified anthocyanin extracts or their respective intestinal digestions.

For instance, the transport efficiency was much higher for the wild pansy and cornflower extracts and their digestions. The main anthocyanin from the cornflower extract is diglycosylated, with both sugar units non-acylated, whereas the main anthocyanin from the wild pansy extract is glycosylated with rutinoside and a glucose that is also non-acylated. The presence of a non-acylated sugar unit may allow these anthocyanins to bind to intestinal glucose transporters, resulting in a higher transport efficiency [8]. Likewise, it has been observed in an in vivo study that anthocyanin rutinosides exhibit a higher concentration in plasma, that is a higher absorption, when compared to anthocyanin glucosides [49]. Conversely, cosmos exhibited the lowest transport efficiency in both cases. The main anthocyanin in the cosmos extract is monoglycosylated; however, this sugar moiety is acylated, rendering this particular anthocyanin incapable of binding to specific intestinal glucose transporters. Altogether, these results align with established literature, which emphasizes the influence of anthocyanin structural arrangement on its intestinal transport [46,50].

Because the physicochemical properties of anthocyanins determine their behavior throughout their digestion and, subsequently, their absorption, it is important to consider these particularities when studying their bioaccessibility and bioavailability [51]. Therefore, these results are of high importance as anthocyanin bioavailability is a critical indicator for evaluating the potential in vivo effects of these bioactive compounds [46].

## 3. Materials and Methods

### 3.1. Plant Materials

Fresh edible flowers from the species *Centaurea cyanus*, *Cosmos bipinnatus*, and *Viola tricolor* were all provided by local producers from both the north and the south of Portugal. The samples were harvested, and about 100 g of petals were treated for the obtention of the anthocyanin-rich extracts in the day after.

### 3.2. Preparation of the Anthocyanin-Rich Extracts

To obtain the anthocyanin-rich extracts, the different edible flowers were submitted to a multiple-step extraction and purification. Briefly, the petals were separated from the flower corollas and the anthocyanins were exhaustively extracted from the former in deionized acidified water with ultrasound-assistance followed by a centrifugation at 13,400× *g* for 10 min and a filtration in 0.22 μm PTFE membrane. The resulting solution was submitted to a liquid-liquid extraction (ethyl acetate/diethyl ether/water, 1:1:1) and the water fraction was collected. The resulting fraction was then submitted to a solid-phase extraction using a reversed-phase C_18_-bounded silica column using water and methanol as solvents to further remove impurities. Finally, to isolate the anthocyanins from other non-charged natural compounds that were not separated by the previous steps, the extract was submitted to a cation-exchange extraction using DSC-MCAX^®^ 1 g/5 mL cartridges. Briefly, the cartridge was conditioned with methanol followed by an equilibration step with 0.1% formic acid aqueous solution. The extracts were applied in a 0.1% formic acid aqueous solution. The non-charged compounds were eluted with methanol and the anthocyanins were eluted with a solution of methanol:dipotassium phosphate 10 mM pH 6.0 (1:1). A dialysis with a 3.5 kDa membrane was performed to remove all the buffer salts and the resulting anthocyanin-rich extracts were freeze-dried and stored at −20 °C until further use.

### 3.3. UHPLC-DAD Analysis

Two chromatographic methods were applied for the different analyses. For the characterization of the anthocyanin-rich extracts from each edible flower, an aliquot of 20 μL was injected and the method was carried out in a Thermo^®^ Finnigan Surveyor Plus HPLC system (Thermo Fisher^®^, Waltham, MA, USA). For the chromatographic separation, conditions were as follows: use of 1% formic acid as solvent A and 1% formic acid/30% acetonitrile as solvent B. The solvent gradient started with 5.5% B up to 64% B (0–7 min), and then up to 80% B (7–10 min). A solution of 100% B was then applied (10–20 min), followed by the equilibration step under the initial conditions (20–25 min). The stationary phase utilized was a reversed-phase C_18_ Hypersil GOLD^TM^ VANQUISH 150 × 2.1 mm with 1.9 μm particle size column with a steady flux of 0.3 mL.min^−^^1^.

For analyses of the stability, simulated digestions, and absorption experiments, the resulting samples were analyzed using a Thermo Scientific^®^ Dionex UltiMate 3000 UHPLC. For each analysis, a 20-μL aliquot was injected under the following conditions: For the chromatographic separation, conditions included use of 5% formic acid as solvent A and 5% formic acid/30% acetonitrile as solvent B. The solvent gradient started with 5.5% B up to 64% B (0–7 min), and then up to 80% B (7–10 min). A solution of 100% B was then applied (10–20 min), followed by the equilibration step under the initial conditions (20–25 min). The stationary phase utilized was a reversed-phase C_18_ Hypersil GOLD^TM^ 50 × 2.1 mm with 1.9 μm particle size column with a steady flux of 0.5 mL.min^−1^.

### 3.4. LC-DAD/ESI-MS Analysis

For the structure analysis of the different anthocyanins present in each edible flower extract, an aliquot of 20 μL was injected for LC-MS analysis in a Thermo Fisher^®^ LTQ XL ion trap quadrupole coupled with a Thermo^®^ Finnigan Surveyor Plus HPLC System. The chromatographic conditions were the same as previously described in Section 3.3. Detection was carried out at 280 nm using a diode array detector (DAD). Double online detection was performed by a photodiode spectrophotometer and mass spectrometry. The mass detector was a Thermo Fisher^®^ LTQ XL ion trap quadrupole equipped with an atmospheric pressure ionization (API) source, using an electrospray ionization (ESI) interface. The vaporizer and the capillary voltages were 5 kV and 4 V, respectively. The capillary temperature was set at 300 °C. Nitrogen was used both as sheath and auxiliary gas at flow rates of 40 and 15, respectively (in arbitrary units). Spectra were recorded in positive ion mode between *m*/*z* 250 and 1500.

### 3.5. Anthocyanin Stability Assays

The edible flower anthocyanin-rich extracts were submitted to anthocyanin stability assays, where three variables were tested: pH values (pH 1, 3, 5, 7, 9, and 11) throughout 0, 15, 30, 45, 60, 75, and 90 min of reaction, at different temperatures (40, 60, 80, and 100 °C). Six solutions of each extract were prepared in a Falcon^®^ tube (Corning^®^, Corning, NY, USA) with Ultrapure MilliQ^®^ Water (Ultrapure water systems, PURELAB^®^ Option Q, Avantor^®^, Radnor, PA, USA) making up a total volume of 2.5 mL, with a final concentration of 0.4 mg/mL. To ensure the integrity of the anthocyanins, each Falcon^®^ tube was acidified with HCl 2M. Recurring to distinct HCl and NaOH solutions, the pH values of each tube were set to 1, 3, 5, 7, 9, and 11, considering a pH error of ±0.1. After waiting 15 min for the pH stabilization, the content of each Falcon^®^ tube was transferred to seven 600 μL Eppendorf^®^ tubes (Eppendorf^®^, Hamburg, Germany), each one correspondent to each time point analyzed. After the transfer, the Eppendorf^®^ tubes for time 0 (controls) were immediately acidified with 1 μL of HCl 6M. Then, the thermoblock was set to the desired temperature (40, 60, 80, or 100 °C) and the Eppendorf^®^ tubes were added with a constant agitation velocity of 400 rpm. At each time point, the respective Eppendorf^®^ tubes were retrieved and immediately acidified with 1 μL of HCl 6M and further submitted to UHPLC-DAD analysis.

### 3.6. Simulated Digestions

Simulated digestions at the oral (oral phase, OP), gastric (gastric phase, GP), and intestinal level (intestinal phase, IP) were performed according to INFOGEST and as described in [52]. In each phase, the anthocyanin-rich extracts, at a concentration of 0.4 mg/mL, were incubated accordingly with simulated fluid electrolyte stock solutions, along with the appropriate digestive enzymes. The prepared solutions were adjusted to pH 7, for the oral and intestinal digestions, and pH 3 for the gastric digestion. All the simulated digestions were performed at a temperature of 37 °C for 10 min (OP), 3 h (GP), and 2 h (IP) in a thermoblock with a constant rotation velocity of 400 rpm. The blank and control solutions were prepared by dilution of the anthocyanin-rich extracts with water or simulated fluid electrolyte stock solutions, respectively. At the end of each simulated digestion, all the tubes were acidified with 1 μL of HCl 6M and further submitted to UHPLC-DAD analysis.

### 3.7. Cell Culture

Human epithelial cell line Caco-2 was grown in monolayers and maintained at 37 °C in an atmosphere of 5% CO_2_ and 90% relative humidity. The cells were routinely maintained in Dulbecco’s Modified Eagle Medium (DMEM) supplemented with 10% fetal bovine serum (FBS) (Sigma^®^ St. Louis, MI, USA, F4135), 1% sodium pyruvate (Sigma^®^ S8636), 1% transferrin (Sigma^®^ T8158), 1% antibiotic/antimycotic solution (100 units/mL of penicillin, 100 mg/mL of streptomycin, and 0.25 mg/mL of amphotericin B) (Sigma^®^ A5955), and 1% L-glutamine (Biowest^®^, Nuaillé, France, X0551-100). The cell culture medium was changed every 2 to 3 days and the cells were kept in exponential growth in T75 flasks. At 90% confluence, a subculture was performed with the harvested cells by trypsinization (0.25%, *w*/*v*, trypsin−EDTA_4_−Na), with a split ratio of 1:3.

### 3.8. MTT Assay

To evaluate the cytotoxicity of the anthocyanin-rich extracts, a MTT assay was performed for the Caco-2 cell line. The cells were seeded at a concentration of 3 × 10^5^ cells/mL, prepared in DMEM, in a 96-well plate. These plates were incubated at 37 °C in an atmosphere of 5% CO_2_ and 90% relative humidity. Throughout 21 days, the cells were routinely maintained as described in Section 3.7. After 21 days of incubation, the cell culture medium was removed, and the wells were washed twice with Hanks’ Balanced Salt solution (HBSS). We prepared a solution of each edible flower’s anthocyanin extract at a concentration of 0.8 mg/mL and a solution of 0.5% dimethyl sulfoxide (DMSO) as a control, both in DMEM. Through serial dilutions, the 96-well plate was incubated with different concentrations of the edible flower extracts for 24 h under the same conditions described above. After the 24 h incubation time, a solution of MTT with a concentration of 0.5 mg/mL was prepared in HBSS. The cells were incubated with the MTT solution and, after the appearance of the purple reaction product, the MTT solution was removed and DMSO was added to solubilize the MTT-derived formazan. After about 20 min, the 96-well plate absorbance was read in a microplate reader (FlexStation^®^ 3 Multi-Mode Microplate Reader) at 570 nm.

### 3.9. Transepithelial Absorption Experiments

Caco-2 cells were plated on 12-well Transwell^®^ polycarbonate permeable inserts, measuring 12 mm in diameter and 0.4 μm in pore size, (Corning Costar, Corning, NY, USA) at 3 × 10^5^ cells/mL. The cells were cultured for 21 days at 37 °C in an atmosphere of 5% CO_2_ and the medium was changed every 2 days. The transepithelial electrical resistance (TEER) was measured using MILLICELL-ERS epithelial voltammeter (Millipore Co., Bedford, MA, USA) with “chopstick” electrodes. The *TEER* of the cell monolayers was calculated according to the following equation, where *R* represents the resistance measured and *A* the surface area of the filter, 1.12 cm^2^):*TEER* (Ω.cm^2^) = *R* × *A*

HBSS with 2% FBS at pH 7.4 was used as a basolateral medium, while on the apical side the anthocyanin extracts or their intestinal digestions were added. The extracts were prepared in HBSS to reach a concentration of 0.8 mg/mL. The digestions were performed as described in 3.6 with an initial concentration of 12.8 mg/mL for the oral digestion. The resulting intestinal digestion samples were submitted to a desalting process with Chromabond^®^ C_18_ 3 mL/500 mg followed by a 1:4 dilution in HBSS before its addition to the Transwell^®^. The transepithelial absorption experiments were performed after 21 days, as the TEER values stabilized around 350–400 Ω.cm^2^. The cell culture medium was removed, and the basolateral side was washed with HBSS with 2% FBS. Then, 0.5 mL of the anthocyanin extracts or their intestinal digestion was added to the apical compartment and 1.5 mL of HBSS with 2% FBS to the basolateral compartment. The transepithelial transport was followed at 37 °C, by taking aliquots from the basolateral side every 15, 30, 60, 120, 180, and 240 min. Every time an aliquot was taken, the same volume of HBSS with 2% FBS was added. Additionally, an aliquot of the apical side content was collected to keep as a control. All the collected aliquots were acidified with 1 μL of HCl 6M and frozen at −18 °C until UHPLC-DAD analysis.

### 3.10. Statistical Analysis

The anthocyanin stability assays were performed at least 3 times (n = 3–6) depending on the flower extract and the results are represented as mean ± SD. The MTT assays were performed at least 3 times (n = 3–9) depending on the flower extract and the results are represented as mean ± SEM. Two-way ANOVA with comparisons between all groups with Tukey’s multiple comparisons test was performed in the case of the simulated digestion experiments (n = 3 for each digestion phase) and in the transepithelial absorption assays (n = 12 and n = 3 for the anthocyanin extracts and their simulated intestinal digestions, respectively). The results are represented as mean ± SEM. All the analyses were performed using GraphPad Prism version 9.5.0 for MacOS, GraphPad Software, San Diego, CA, USA, www.graphpad.com. * *p* < 0.05; ** *p* < 0.01; *** *p* < 0.001; **** *p* < 0.001.

## 4. Conclusions

Edible flowers are increasingly considered a relevant alternative food source for the near future. Particularly, anthocyanin-rich edible flowers present a valuable source of complex anthocyanins with highly variable glycosylation and acylation patterns. Wild pansy, cosmos, and cornflower exhibited both mono- and polyglycosylated anthocyanins. Anthocyanins were all glycosylated with rutinoside for wild pansy, monoglycosylated with glucose for cosmos, and diglycosylated for cornflower. Overall, the rutinoside and monoglucoside anthocyanin extracts from wild pansy and cosmos were less stable under all tested pH and temperature conditions over time, and under the distinct digestive processes in the gastrointestinal tract. From a nutritional and food processing perspective, to make the most out edible flowers’ anthocyanin content, the results herein suggest that they should be minimally processed, as they are mostly stable at lower temperatures and pH values. From a technological point of view, this is crucial information for the production of novel functional foods derived from anthocyanin-rich edible flowers. Nevertheless, it is important to notice that these negative effects were also time-dependent, so it is crucial for industry to assess these parameters together when choosing strategies to follow.

Additionally, the rutinoside and diglucoside anthocyanin extracts from wild pansy and cornflower exhibited the highest absorption efficiencies at the small intestine level. From a molecular mechanistic perspective, it is known that glucose transporters are one of the main mechanisms from which anthocyanins are absorbed at the gastrointestinal level. In previous studies, it was shown that anthocyanins with free sugar moieties present a higher absorption in gastrointestinal cell models [22,53]. In this case, anthocyanins from both wild pansy and cornflower present a free sugar moiety at position C5.

Overall, cornflower exhibited both high chemical stability and intestinal absorption efficiencies. Regarding the anthocyanin content and behavior in these different approaches, we can rank the edible flowers in the following order: wild pansy < cosmos < cornflower.

Altogether, the results suggest that the overall health outcomes of edible flower anthocyanins depends on their parental structure. Therefore, when ingesting such types of foods, it is important not only to pay attention to the general food processing conditions, but also to the type/structure of the bioactives it contains to obtain the best nutritional and health outcomes.

Edible flowers present a promising alternative food source, capable of enhancing both culinary experiences and the nutritional content of meals. Whether consumed as a whole or as extracts incorporated into functional foods, these cultivars contribute unique flavors, textures, and potentially beneficial compounds. However, it is crucial to exercise caution as certain edible flowers may harbor potential toxic molecules that can pose risks to human health. Examples of such compounds include pyrrolizidine alkaloids or triterpene alcohols, in certain varieties. A recent study reviewed investigation of the use of different edible flowers picked from the wild and cultivated in private gardens or market gardens and assessed the presence of potential toxic compounds. From a total of 23 edible flowers, 9 were shown to have compounds with toxic or potentially toxic effects, including well-known species such as *Borago officinalis* L., *Syringa vulgaris* L., and *Tropaeolum majus* L., in different parts of the plants [54].

A lack of botanical knowledge among the general population and the assumption that natural products do not pose any risk to health turn into a potentially dangerous combination of factors that can lead to the consumption of dangerous species. For some flowers, projections have been made by EFSA and JECFA about the total daily amounts, including *Tropaeolum majus* L. (39.5 g), *Achillea millefolium* L. (18 g), and *Galium odoratum* L. (7 g).

Therefore, to ensure the safe integration of edible flowers into diets, thorough assessments and an awareness of potential toxins are imperative, emphasizing the importance of responsible consumption practices and careful consideration before widespread adoption.

Also, for the incorporation of these edible flower anthocyanin extracts into novel food sources, attention should be paid to the type of extraction used, which should be based on number should be provided. Please check the accuracy of funding data and any other information carefully green extraction methodologies. For this purpose, it is highly preferable to use full extracts (that are not only free of non-water solvents but also significantly cheaper), while the purified extracts are crucial for mechanistic and fundamental studies.

## Figures and Tables

**Figure 1 pharmaceuticals-17-00191-f001:**
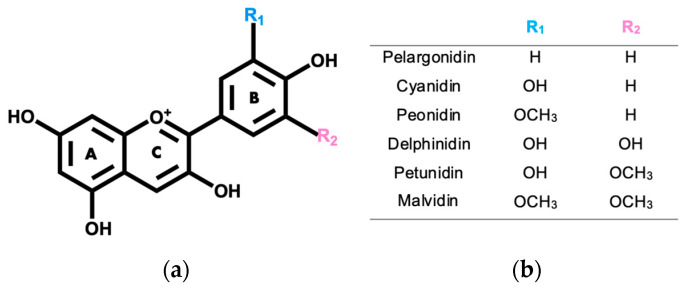
(**a**) Anthocyanidin base structure. (**b**) Substituents for R_1_ and R_2_ groups for the most abundant anthocyanidins in edible flowers.

**Figure 2 pharmaceuticals-17-00191-f002:**
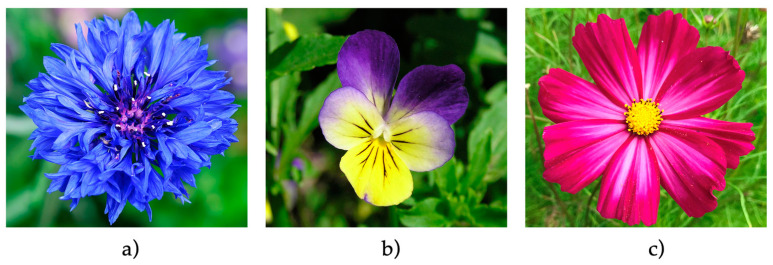
Examples of some of the most widely consumed anthocyanin-rich edible flowers. (**a**) Cornflower (*Centaurea cyanus*) (Source: https://www.flickr.com/photos/27656017@N02/ (accessed on 16 December 2023), licensed under https://creativecommons.org/licenses/by-sa/2.0/?ref=openverse (accessed on 16 December 2023)). (**b**) Wild pansy (*Viola tricolor*) (Source: https://commons.wikimedia.org/w/index.php?title=User:Tom.k&action=edit&redlink=1 (accessed on 16 December 2023), licensed under https://creativecommons.org/licenses/by-sa/2.5/?ref=openverse (accessed on 16 December 2023)). (**c**) Cosmos (*Cosmos bipinnatus*) (Source: https://www.flickr.com/photos/93416311@N00 (accessed on 16 December 2023), licensed under https://creativecommons.org/licenses/by/2.0/?ref=openverse (accessed on 16 December 2023)).

**Figure 3 pharmaceuticals-17-00191-f003:**
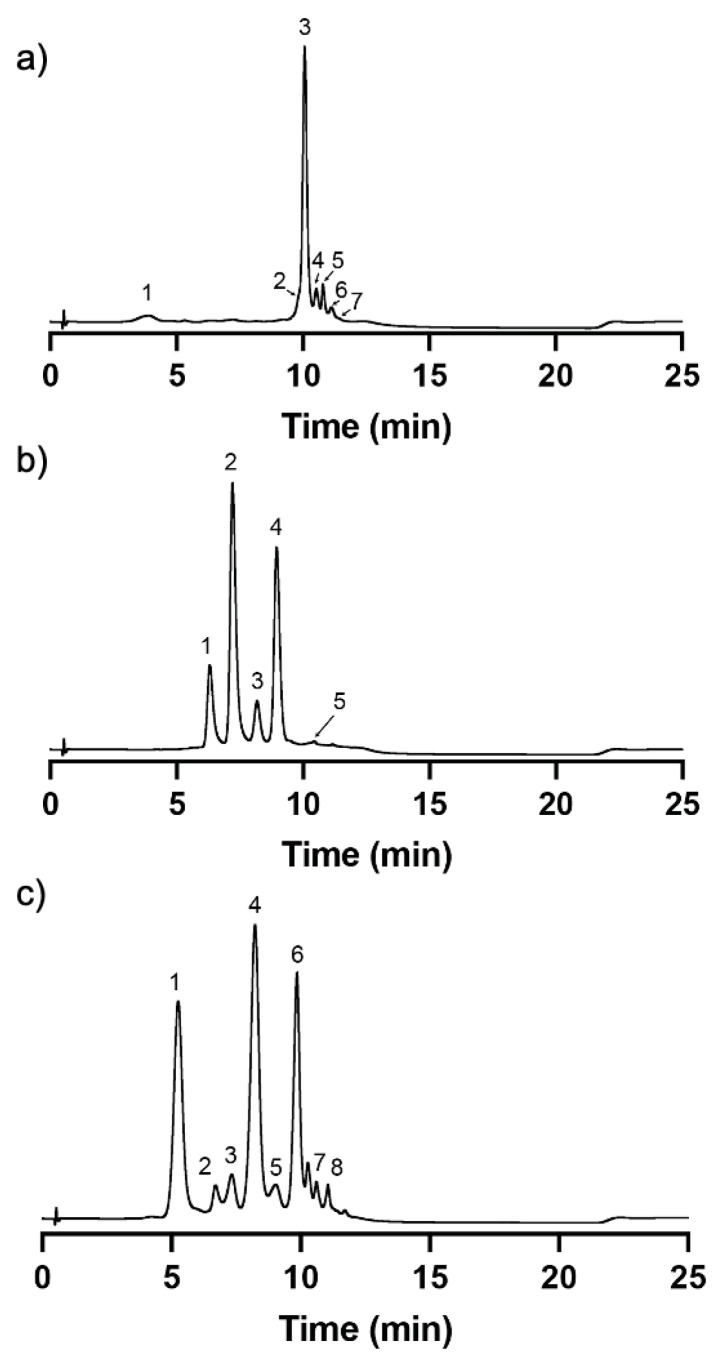
UHPLC-DAD chromatograms of the different purified extracts from (**a**) wild pansy, (**b**) cosmos, and (**c**) cornflower, recorded at 280 nm. Identification of the peaks can be found in Table 1, Table 2 and Table 3, respectively.

**Figure 4 pharmaceuticals-17-00191-f004:**
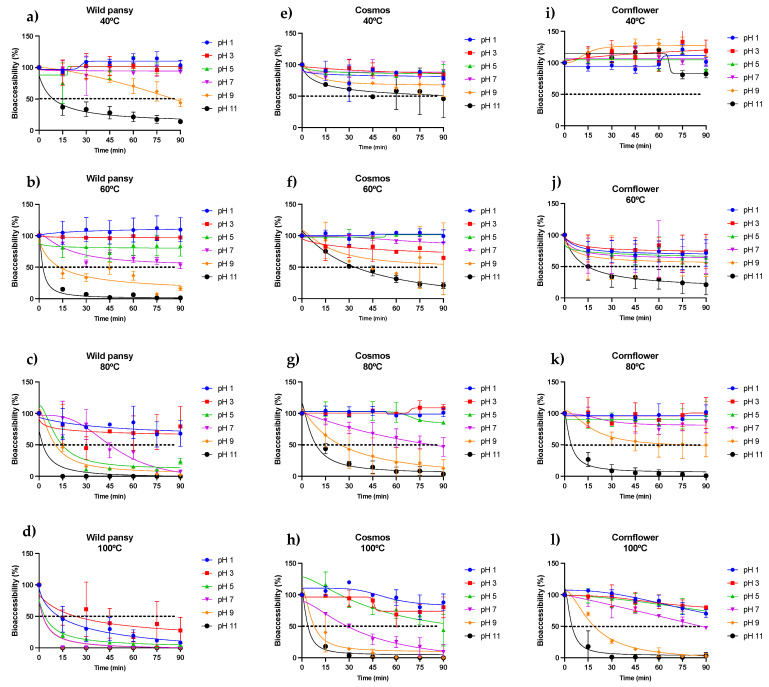
Temperature and pH effect on the bioaccessibility of the anthocyanins present in each purified extract from wild pansy (**a**–**d**), cosmos (**e**–**h**), and cornflower (**i**–**l**). Each experiment was repeated at least 3 times (*n* = 3–6). A non-linear regression was performed to obtain a curve fitting of the points and a dotted line was traced at 50% for a better visualization and interpretation of the results. The results are presented as mean ± SD.

**Figure 5 pharmaceuticals-17-00191-f005:**
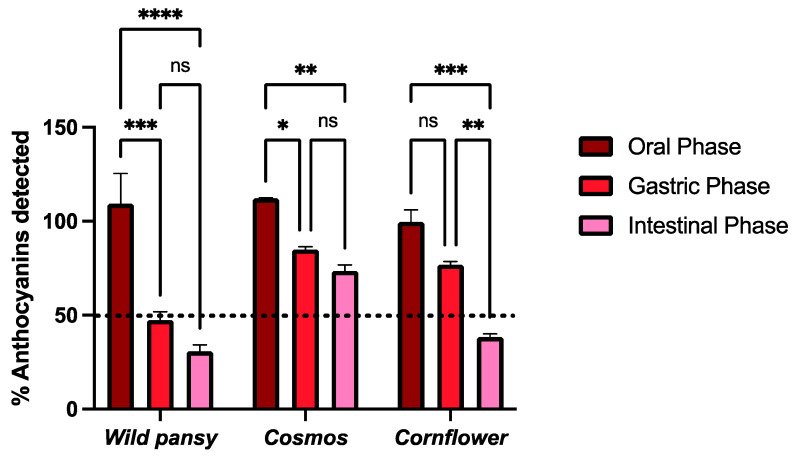
The effect on the total anthocyanin content after simulated digestions at oral, gastric, and intestinal levels for the anthocyanin-rich extracts of the selected edible flowers. The results are expressed as total amount of anthocyanins detected (%) using UHPLC analysis (mean ± SEM). The statistical analysis was performed using a two-way analysis of variance with Tukey’s multiple comparison test. Each experiment was performed three times (*n* = 3). * *p* < 0.05; ** *p* < 0.01; *** *p* < 0.001; **** *p* > 0.0001; ns—non significant.

**Figure 6 pharmaceuticals-17-00191-f006:**
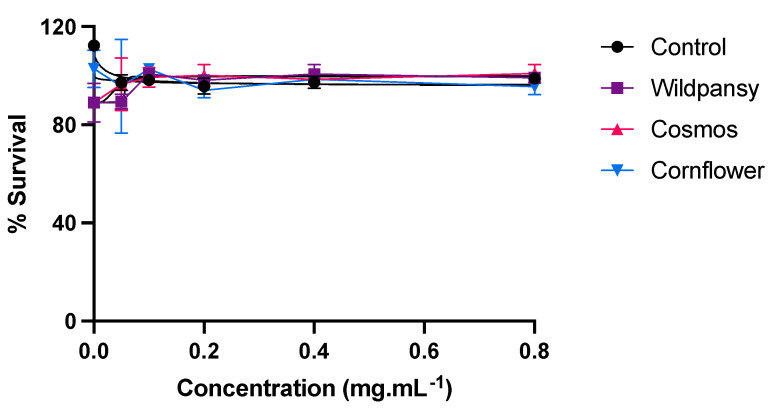
Cytotoxicity of the edible flowers’ purified anthocyanin extracts on a Caco-2 cell line, evaluated by the MTT assay. The cells were treated with the extracts for 24 h, with concentrations up to 0.8 mg/mL. Each experiment was performed at least three times (*n* = 3–9). The results are presented as mean ± SEM.

**Figure 7 pharmaceuticals-17-00191-f007:**
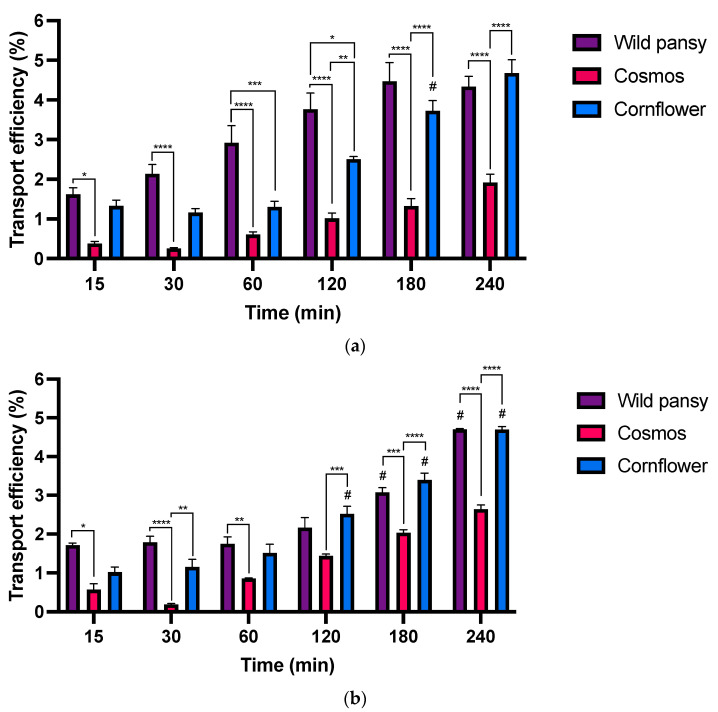
(**a**) Absorption efficiency of the different anthocyanin extracts up to 4 h in the intestinal model using Caco-2 cells with Transwell^®^ systems. (**b**) Absorption efficiency of the different anthocyanin extracts’ intestinal digestions up to 4 h in the intestinal model using Caco-2 cells with Transwell^®^ systems. Each experiment was performed at least three times (*n* = 3–12). #: Significantly different from the previous time point for the same EF, for a *p* < 0.05 *, *p* < 0.05; ** *p* < 0.01; *** *p* < 0.001; **** *p* < 0.0001.

**Table 1 pharmaceuticals-17-00191-t001:** UHPLC-DAD-ESI/MS-MS identification, on positive mode, of the anthocyanins identified in wild pansy at 280 nm. Retention time (R_t_), wavelengths of maximum absorption (λ_max_), mass spectral data, and tentative identification are represented.

Peak	R_t_ (min)	λ_max_ (nm)	ESI Full MS (*m*/*z*)	ESI full MS^2^ (*m*/*z*)	Tentative Identification
**1**	4.41	277, 520	919.18	757.27; 303.15	Delphinidin-3-(4′-cis-p-coumaroyl)-O-rutinoside-5-O-glucoside
**2**	9.87	282, 526	773.13	627.12; 465.15; 303.07	Delphinidin-3-O-rutinoside-5-O-glucoside
**3**	10.09	280, 529	919.33	757.25; 465.13; 303.12	Delphinidin-3-(4′-trans-p-coumaroyl)-O-rutinoside-5-O-glucoside
**4**	10.68	253, 523	757.18	611.13; 465.18; 303.10	Delphinidin-3-(4′-cis-p-coumaroyl)-O-rutinoside
**5**	10.84	280, 313, 529	1081.27	756.80; 626.80	Delphinidin-3-(4′-trans-p-coumaroyl)-O-rutinoside-5-O-(6′-caffeoyl)-glucoside
**6**	11.31	282, 523	903.36	741.26; 449.20; 287.12	Cyanidin-3-(4′-cis-p-coumaroyl)-O-rutinoside-5-O-glucoside
**7**	11.86	280, 532	757.34	465.19; 303.10	Delphinidin-3-(4′-trans-p-coumaroyl)-O-rutinoside

**Table 2 pharmaceuticals-17-00191-t002:** UHPLC-DAD-ESI/MS-MS identification, on positive mode, of the main anthocyanins identified in cosmos at 280 nm. Retention time (R_t_), wavelengths of maximum absorption (λ_max_), mass spectral data, and tentative identification are represented.

Peak	R_t_ (min)	λ_max_ (nm)	ESI Full MS (*m*/*z*)	ESI Full MS^2^ (*m*/*z*)	Tentative Identification
**1**	6.36	280, 514	449.19	287.08	Cyanidin 3-O-glucoside
**2**	7.23	280, 517	595.32	449.17; 287.08	Cyanidin 3-O-(6″-p-coumaroyl-glucoside)
**3**	8.26	279, 517	463.23	301.08	Peonidin 3-O-glucoside
**4**	8.98	280, 518	609.28	463.18; 301.11	Peonidin 3-O-(6″-p-coumaroyl-glucoside)
**5**	10.86	280, 339, 523	611.1	465.08	Delphinidin 3-O-(6″-p-coumaroyl-glucoside)

**Table 3 pharmaceuticals-17-00191-t003:** UHPLC-DAD-ESI/MS-MS identification, on positive mode, of the main phenolic compounds identified in cornflower at 280 nm. Retention time (R_t_), wavelengths of maximum absorption (λ_max_), mass spectral data, and tentative identification are represented.

Peak	R_t_ (min)	λ_max_ (nm)	ESI Full MS (*m*/*z*)	ESI Full MS^2^ (*m*/*z*)	Tentative Identification
1	5.29	277, 511	611.31	449.19; 287.15	Cyanidin-3,5-O-diglucoside
2	7.15	280, 328, 514	873.38	711.34; 611.29; 549.26; 287.11	Cyanidin 3-O-(6″-succinylglucoside)-sophoroside
3	7.43	277, 514	697.29	535.25; 449.23; 287.10	Cyanidin-3-O-(6″-malonylglucoside)-5-O-glucoside isomer 1
4	8.27	277, 514	697.34	535.22; 287.15	Cyanidin-3-O-(6″-malonylglucoside)-5-O-glucoside isomer 2
5	9.19	280, 514	449.24	287.1	Cyanidin-3-O-glucoside
6	9.89	277, 514	711.38	549.23; 287.15	Cyanidin-3-O-(6″-succinylglucoside)-5-O-glucoside
7	11.04	268, 517	595.27	433.22; 271.13	Pelargonidin-3,5-O-diglucoside
8	11.12	274, 517	549.25	287.11	Cyanidin 3-O-(6″-succinyl-glucoside)

## Data Availability

Data is contained within the article.

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
