# Peer review of "First Insights on the Bioaccessibility and Absorption of Anthocyanins from Edible Flowers: Wild Pansy, Cosmos, and Cornflower"

_pharmaceuticals, 2024, doi:10.3390/ph17020191_

Round 1

Reviewer 1 Report

Comments and Suggestions for Authors

I send a review of manuscript ID number pharmaceuticals-2842820, of the authors: Margarida Teixeira, Lorenzo De Luca, Ana Faria, Matteo Bordiga, Victor de Freitas, Nuno Mateus, Hélder OliveiraFirst insights on the bioaccessibility and absorption of anthocyanins from edible flowers: wild pansy, cosmos and cornflower.

I think that the manuscript concerns an interesting line of scientific research related to the determination of the stability of anthocyanins found in edible flowers (cornflower, wild pansy and cosmos) subjected to different food processing conditions, as well as the analysis of their bioavailability at the intestinal level. I also believe that, this manuscript is relevant to the publication in Pharmaceuticals, but the authors should make a minor revision.

  1. Results and discussion:

Generally in this part of manuscript please align the text to the right margin.

Page 3 & 4, lines 111-150; I would ask the authors to write this part of the paper more in the form of a discussion rather than a description as in the introduction

Page 4, line 164  - …was despicable?

Page 8, lines 256-259; (Figure 4) - Please provide information on the standard deviation (±SD) and number of repetitions (n=) in the appropriate place under the Figure 4. Please provide information on standard deviation (±SD) and number of replicates (n =) in the appropriate space under Figure 4. This information should also be provided by the authors in the Materials and methods section of the manuscript - 3.10. Statistical analysis.

Page 8, line 273 - Please add information regarding the Figure or Table on which the discussed research results are included. Please also include such information (or, more often, stir it up) in the following discussion.

  1. Materials and methods:

Page 13, lines 453-456; How much of each fresh edible flowers sample was taken for testing?

Page 16, lines 604-608; In this section, please provide information on how many replicates were performed in the submitted studies. Please provide information on standard deviation (±SD) and number of replicates (n =) .

4.     Conclusions

Page 16, lines 611-615; I think that the conclusions are too general (laconic) and written too briefly, and do not fully capture the essence of the wide range of research carried out by the Authors. Please indicate the most suitable plant (among the fresh, edible flowers tested), pH conditions, temperature, ect.

References

All literature is cited in the text.

Author Response

Dear reviewer we thank you for your suggestions to improve the manuscript. 

Please find below all the assessments in made:

“Results and discussion: Generally in this part of manuscript please align the text to the right margin.”

Thank you for your relevant suggestion. All the text was aligned accordingly.

“Page 3 & 4, lines 111-150; I would ask the authors to write this part of the paper more in the form of a discussion rather than a description as in the introduction”

Once again, thank you for your suggestion. The authors found relevant to have an explanatory introduction in the section (2.1) that we find to be crucial to give context for the further experimental approach. Nevertheless, we made an effort to improve the discussion of the section by adding a higher degree of detail about the interaction processes that occur with between the compounds and the cation-exchange chromatography functional groups during the process in itself.

“Page 4, line 164  - …was despicable?”

We agree that the term was not the more appropriate. Therefore, the sentence that finished with “… was despicable” was corrected to “… the presence of other compounds rather than anthocyanins was undectable, confirming the high purity degree of the different extracts”.

“Page 8, lines 256-259; (Figure 4) - Please provide information on standard deviation (±SD) and number of replicates (n =) in the appropriate space under Figure 4. This information should also be provided by the authors in the Materials and methods section of the manuscript - 3.10. Statistical analysis.”

Thank you for you relevant information. Indeed, this is a crucial point that was missed. Both the number of replicates (n) and standard deviation (SD) are now provided within the figure captions and in the materials and methods section on the subtopic 3.10. Statistical analysis.

“Page 8, line 273 - Please add information regarding the Figure or Table on which the discussed research results are included. Please also include such information (or, more often, stir it up) in the following discussion.”

A reformulation of figure 4 was done due to the high number of results presented on it. Several letters were added to identify the different results regarding temperature of flower species. This helped us to properly identify them during the discussion of this section.

“Materials and methods: Page 13, lines 453-456; How much of each fresh edible flowers sample was taken for testing?”

The amount of  edible flower sample used to make the extract of anthocyanins is now stated on section 3.1.

“4.     Conclusions: Page 16, lines 611-615; I think that the conclusions are too general (laconic) and written too briefly, and do not fully capture the essence of the wide range of research carried out by the Authors. Please indicate the most suitable plant (among the fresh, edible flowers tested), pH conditions, temperature, ect.”

Thank you for the suggestion. The conclusion section was fully edited and is now more elaborated and detailed, giving more emphasis to the main results obtained in the different experimental procedures.

Reviewer 2 Report

Comments and Suggestions for Authors

Dear Authors,

 The interest in the valorization of bioactive compounds from edible plants has become increasingly greater, therefore this manuscript brings added value to the study of anthocyanins from these sources. The presented manuscript is well structured, synthesized and elaborated. The extraction methods are quite laborious and worth applying, in my opinion, if you want to obtain anthocyanin supplements, considering their low bioavailability. I would have some recommendations: 1. The quantification of total anthocyanins, according to which their bioavailability was studied, does not appear anywhere in the manuscript. 2. In section 2.3.2. it would be of interest to specify in the text the bioaccessibility of anthocyanins after oral, gastric and intestinal digestion, as in the case of temperature, for example. 3. The conclusions are very few and general. Considering the fact that many data and results were presented, the conclusions should be more elaborated and more precise.    

Author Response

Dear reviewer we thank you for your suggestions to improve the manuscript. 

Please find below all the assessments to your questions: 

“1. The quantification of total anthocyanins, according to which their bioavailability was studied, does not appear anywhere in the manuscript.”

Thank you for your observation. The main aim of the study was to evaluate the bioaccessibility and intestinal absorption of the anthocyanins present in the edible flowers by testing their different extracts. Thus, it was made a direct comparison (using the same extract concentrations) so that the potential structural differences were evidenced. Considering this, the quantification of total anthocyanins was not the scope of the present article, therefore at this point we did not perform such quantification.

“2. In section 2.3.2. it would be of interest to specify in the text the bioaccessibility of anthocyanins after oral, gastric and intestinal digestion, as in the case of temperature, for example.”

Once again thank you for the relevant suggestion. A quantitative analysis of the results in section 2.3.2. was added to the main discussion text.

“3. The conclusions are very few and general. Considering the fact that many data and results were presented, the conclusions should be more elaborated and more precise.” 

Thank you for the suggestion. The conclusion section was fully edited and is now more elaborated and detailed, giving more emphasis to the main results obtained in the different experimental procedures.

Reviewer 3 Report

Comments and Suggestions for Authors

In the near future, edible flowers may become an important alternative food source. There is growing evidence of their health benefits. Taking this into account, studies were conducted on the behavior of anthocyanin-rich extracts from selected edible flowers: pansy, cosmos and cornflower under various food processing conditions and after ingestion using digestion modeling, as well as their absorption at the intestinal level. The obtained results provide the first insight into the bioavailability and absorption of anthocyanins from these edible flowers, as well as the potential outcomes of using such alternative food sources. In general, the extracts of thocyanins monoglucoside and rutinoside anthocyanins extracts were found to be less stable under different conditions of pH, temperature and time, and under different digestive processes in the gastrointestinal tract. However, extracts of diglucoside and rutinoside anthocyanins showed the highest absorption efficiency at the intestinal level. A number of other interesting results are reported in this article.

In general, the article is well written and well designed and only minor editing of English language required. The obtained results are interesting and can be published. In my opinion, the disadvantage of the article is that the conclusions section is very short and written in general phrases. In the conclusions, it would be appropriate to write in more detail and provide the main specific results obtained in these studies.

Comments on the Quality of English Language

Minor editing of English language required

Author Response

Dear reviewer thank you for your suggestions to improve our manuscript.

Please check below all the assessments to your questions: 

“In general, the article is well written and well designed and only minor editing of English language required. The obtained results are interesting and can be published. In my opinion, the disadvantage of the article is that the conclusions section is very short and written in general phrases. In the conclusions, it would be appropriate to write in more detail and provide the main specific results obtained in these studies.”

Thank you for the suggestion. The conclusion section was fully edited and is now more elaborated and detailed, giving more emphasis to the main results obtained in the different experimental procedures.

Reviewer 4 Report

Comments and Suggestions for Authors

The manuscript entitled “First insights on the bioaccessibility and absorption of anthocyanins from edible flowers: wild pansy, cosmos and cornflower” investigated the bioaccessibility and absorption of anthocyanins in the selected edible flowers.

The results are interesting with the novelty of the research object, and the research questions are clear and well-resolved. However, the authors should consider the following issues:

The abstract was written in very general expression. The authors should revise comprehensively by adding more details about the results (with numerical data) and highlighting new findings of the current study.

A more suitable explanation should be provided for the experiment on the stability of anthocyanins at different pH levels. If the author intends to offer a recommendation for the most suitable pH for storing or using anthocyanin products within the scope of this study, the author should assess the samples for each pH treatment through bioaccessibility, absorption, and cytotoxic assays. Consequently, the pH that yields the highest bioaccessibility and absorption of those anthocyanins should be concluded, rather than conducting separate screenings as currently done.

In addition, the authors should also discuss the precautions when using unsafe chemicals in sample extraction, pH adjustment, as well as the risks associated with undesired substances and impurities in the process of synthesizing and preserving natural products. These factors may pose health risks such as liver and nervous system toxicity. The following article could serve as a reference: 10.3390/ijms21145011.

The iThenticate software detected a high level of similarity in the following sections: L65-67, L504-511, L545-553, L575-582. The authors should paraphrase them.

The conclusion was too general. The authors should improve it.

In summary, I recommend a major revision for this manuscript.

Comments on the Quality of English Language

Moderate editing of English language required

Author Response

Dear reviewer thank you for your suggestions for the improvement of our manuscript. 

Below you can find all the assessments to your questions: 

“The abstract was written in very general expression. The authors should revise comprehensively by adding more details about the results (with numerical data) and highlighting new findings of the current study.”

The abstract was enhanced, providing more emphasis to the results obtained (while also mentioning some relevant numerical data) and the main findings from the current study.

“A more suitable explanation should be provided for the experiment on the stability of anthocyanins at different pH levels. If the author intends to offer a recommendation for the most suitable pH for storing or using anthocyanin products within the scope of this study, the author should assess the samples for each pH treatment through bioaccessibility, absorption, and cytotoxic assays. Consequently, the pH that yields the highest bioaccessibility and absorption of those anthocyanins should be concluded, rather than conducting separate screenings as currently done.”

It is well known that the anthocyanin chemical stability highly depends on pH. Our aim by testing a wide range of pH values (ranging from highly acidic to basic pH values) in the anthocyanin stability assays was to perform an initial screening of the physicochemical properties of the anthocyanins from each extract. This allowed us to unveil which temperature and pH values were more suitable to maintain the anthocyanin chemical stability. This gives us important insights on the behavior of each edible flower anthocyanin extract when submitted to different cooking techniques/food processing methods (such idea is now stated in the main text). The next results dealing with the digestion effect at the different gastrointestinal cavities were performed according only to relevant pH values such as pH 3 for the oral and intestinal phase and pH 7 for the gastric phase, as it is stipulated by INFOGEST standardized protocol. Regarding the absorption studies, the same idea was followed, therefore the selected pH (7.4) was to mimic the in vivo conditions in the small intestine.

“In addition, the authors should also discuss the precautions when using unsafe chemicals in sample extraction, pH adjustment, as well as the risks associated with undesired substances and impurities in the process of synthesizing and preserving natural products. These factors may pose health risks such as liver and nervous system toxicity. The following article could serve as a reference: 10.3390/ijms21145011.”

Thank you for the observation. Regarding the use of unsafe solvents during the edible flowers anthocyanin extraction and purification: In this case, the aim was to understand how this type of phytochemicals is absorbed at the intestinal level (the main site for anthocyanin absorption). From a nutritional perspective, the consumer would be ingesting the whole flower (which besides anthocyanins contains other classes of phytochemicals) and not the anthocyanin extract obtained through the several extractions/purification steps. Besides, all edible flowers under study do not possess any kind of significant toxicity and/or anti-nutritional substances which could potentially exert hepato- and neurotoxicity. As far as the extracts incorporation in food products, these particularities should be taken into consideration (which is now stated on the manuscript); however, that is not the final aim of the present study. For the extraction process it is used, when possible, green solvents (such as water for the anthocyanin extraction from the petals of the edible flowers). The organic solvents used for the purification techniques (such as liquid-liquid extraction, reversed phase C18 and cation-exchange chromatography) were all evaporated from the final sample, so they do not interfere with the potential toxicity of the final purified extract in the used cell lines.

“The iThenticate software detected a high level of similarity in the following sections: L65-67, L504-511, L545-553, L575-582. The authors should paraphrase them.”

Regarding the rephrase suggestions, the sentence on the introduction was rewritten. The rest of the sections in materials and methods represent standardized protocols from articles previously published.

“The conclusion was too general. The authors should improve it.”

Thank you for the suggestion. The conclusion section was fully edited and is now more elaborated and detailed, giving more emphasis to the main results obtained in the different experimental procedures.

Round 2

Reviewer 4 Report

Comments and Suggestions for Authors

The author has addressed the majority of the issues raised.

However, I still do not agree with the way the explanation and assertion have been made that the extracts in this study do not contain potentially toxic substances unless the author conducts any tests on this.

It should be remembered that potentially toxic substances in plants are many times more toxic than in animals. The difference lies only in the dose absorbed and the risk of accumulation in the body if not excreted. Discussing potential toxicity is necessary in studies of natural products and functional foods.

Comments on the Quality of English Language

Minor editing of English language required

Author Response

"However, I still do not agree with the way the explanation and assertion have been made that the extracts in this study do not contain potentially toxic substances unless the author conducts any tests on this.

It should be remembered that potentially toxic substances in plants are many times more toxic than in animals. The difference lies only in the dose absorbed and the risk of accumulation in the body if not excreted. Discussing potential toxicity is necessary in studies of natural products and functional foods."

Dear reviewer, thank you for your new input. Indeed we may misunderstood your first inquiry regarding the potential toxicity of the edible flowers herein tested. We have include a new section on the conclusions assessing this issue related with the potential presence of compounds that may present some level of toxicity to humans including for instance the pyrrolizidine alkaloids.

The newly added section is highlighted in red.

We hope that now we have met your insightful suggestion.